

# Tensiomyographical responsiveness to peripheral fatigue in quadriceps femoris

Rodrigo Martín-San Agustín[1], Francesc Medina-Mirapeix[2], José Casaña-Granell[1], José A. García-Vidal[2], Carmen Lillo-Navarro[3] and Josep C. Benítez-Martínez[1]

[1] Department of Physiotherapy, University of Valencia, Valencia, España
[2] Department of Physiotherapy, University of Murcia, Murcia, Spain
[3] Department of Pathology and Surgery, University Miguel Hernández, San Joan, Spain

## ABSTRACT

**Background**. Fatigue influences athletic performance and can also increase the risk of injury in sports, and most of the methods to evaluate it require an additional voluntary effort. Tensiomyography (TMG), which uses electrical stimulation and a displacement sensor to evaluate muscle contraction properties of one or more muscle bellies, has emerged as a technique that can assess the presence of peripheral and central fatigue without requiring additional voluntary efforts. However, the evaluation of the TMG's ability to detect fatigue is limited, both at the level of muscle bellies and statistical methods. Thus, the aim of the present study was twofold: (i) to examine and compare the tensiomyographical responsiveness to quadriceps femoris (QF) fatigue by multiple statistical methods and (ii) to analyze sex differences in the variation produced by fatigue in TMG parameters.

**Methods**. Thirty-nine recreational athletes participated (19 males/20 females; aged $22 \pm 2$ years). TMG parameters of QF bellies and maximal voluntary isometric contraction (MVIC) were measured before and after a fatigue protocol. TMG parameters used were maximum radial deformation (Dm), contraction time between 10–90% of the Dm (Tc), contraction velocity between 10–90% (Vc) and of the first 10% (V10) of the Dm. Internal responsiveness of TMG to fatigue was analyzed by paired t-test and standardized response mean (SRM). External responsiveness was examined by correlations, regression models, and receiver operating characteristic (ROC) curves.

**Results**. All TMG parameters, except for Tc of rectus femoris and vastus medialis, showed large internal responsiveness. In adjusted regression models by sex, only Dm and V10 of rectus femoris were statistically associated ($p < 0.05$) with b coefficients of 0.40 and 0.43, respectively. r2 explained the 22% of the total variance. In addition, these parameters could discriminate between QF with and without fatigue.

**Conclusion**. Since the QF is the main strength contributor during multiple physical activities, clinicians and trainers will be able to discriminate the presence of fatigue and the magnitude of changes in the QF strength by TMG evaluation.

Corresponding author
Francesc Medina-Mirapeix,
mirapeix@um.es

## INTRODUCTION

Fatigue is defined as a decline in muscular performance which produces a reduction in strength and power generation (*Ditroilo et al., 2011*). It can be further explained by

factors related to the central nervous system as changes at the spinal level (*Gandevia, 2001*) or by peripheral factors associated to the muscle, such as failure of transmission at the neuromuscular junction (*Allen, Lamb & Westerblad, 2008*). Its manifestation can vary in subjects with different training backgrounds (*Garrandes et al., 2007*), type of muscle contraction performed (*Kay et al., 2000*), or even between sex (*Albert et al., 2006*; *Martin & Rattey, 2007*; *Ansdell et al., 2017*).

Since fatigue influences athletic performance (*Thorlund et al., 2008*; *Ditroilo et al., 2011*) and can also increase the risk of injury in sports (*Zebis et al., 2011*; *Liederbach et al., 2014*), its study has been of interest. Multiple methods have been used to induce fatigue, both central fatigue in several muscle groups or peripheral fatigue in a specific muscle (*García-Manso et al., 2011*; *Hunter et al., 2012*; *Macgregor et al., 2016*; *Wiewelhove et al., 2017*; *Wiewelhove et al., 2018*). Thus, fatigue has been evaluated after short term (*Macgregor et al., 2016*; *Abelairas-Gómez et al., 2018*) and long duration efforts, such as several days of intense training sessions (*Wiewelhove et al., 2017*), and also after isolated long sessions (2–12 h approximately) (*Lepers et al., 2002*; *García-Manso et al., 2011*; *Wiewelhove et al., 2018*).

The most used fatigue evaluation methods have been based on changes in maximal voluntary isometric contractions (MVICs) (*Lepers et al., 2002*; *Zebis et al., 2011*), muscle activation (*Garrandes et al., 2007*; *Thorlund et al., 2008*), kinematics and kinetics measurements (*Liederbach et al., 2014*; *Tam et al., 2017*), biochemical markers (*Gorostiaga et al., 2012*), or muscular contractile properties (*García-Manso et al., 2011*; *De Paula Simola et al., 2016*). In a situation of fatigue, most of these methods would require an additional voluntary effort. Their application therefore would not be practical or safe facing the possible presence of central inhibition (*Graven-Nielsen et al., 2002*), or the possibility of increase any extant muscular damage (*Macgregor et al., 2016*).

Tensiomyography (TMG), which uses electrical stimulation and a displacement sensor to evaluate muscle contraction properties of one or more muscle bellies (*Valencic & Knez, 1997*), has emerged as a technique that can assess the presence of peripheral and central fatigue without requiring additional voluntary efforts (*García-Manso et al., 2011*; *De Paula Simola et al., 2016*). Peripheral fatigue has been evaluated by TMG for specific muscle group from both lower and upper limbs (*Carrasco et al., 2011*; *Hunter et al., 2012*; *García-Manso et al., 2012*; *Macgregor et al., 2016*). In contrast, central fatigue has been evaluated only in the lower limb, being quadriceps femoris (QF) the most studied muscle group (*García-Manso et al., 2011*; *De Paula Simola et al., 2015*; *De Paula Simola et al., 2016*; *Giovanelli et al., 2016*; *Raeder et al., 2016*; *Wiewelhove et al., 2017*).

Responsiveness is defined as the ability of a tool to detect important clinical changes over time (*Guyatt et al., 1989*). Since this characteristic is essential to assess fatigue by TMG, it has been analyzed by multiple studies (*García-Manso et al., 2011*; *Hunter et al., 2012*; *De Paula Simola et al., 2015*; *De Paula Simola et al., 2016*; *Giovanelli et al., 2016*; *Macgregor et al., 2016*; *Raeder et al., 2016*; *Wiewelhove et al., 2017*; *Abelairas-Gómez et al., 2018*). Most of these studies evaluated one muscle belly and they used one or two statistical methods of either internal responsiveness (e.g., paired $t$-test and effect size) or external responsiveness (correlation with reference measure or regression models) Internal responsiveness is the ability of a measure to change over a set period and external

responsiveness reflects the extent to which changes in a measure over a specified time frame related to corresponding changes in an external reference measure of health status (*Husted et al., 2000*). Overall, TMG of those evaluated muscle bellies has shown to be internally and externally responsive in assessing central fatigue (*García-Manso et al., 2011*; *De Paula Simola et al., 2015*; *De Paula Simola et al., 2016*; *Giovanelli et al., 2016*; *Raeder et al., 2016*; *Wiewelhove et al., 2017*), and internally responsive to peripheral fatigue (*Hunter et al., 2012*; *García-Manso et al., 2012*; *Macgregor et al., 2016*; *Abelairas-Gómez et al., 2018*). However, to the best of our knowledge, the external responsiveness of TMG has not been yet assessed for peripheral fatigue, and therefore comparisons between internal and external responsiveness has not been established. Furthermore, to our knowledge, TMG responsiveness has not been simultaneously evaluated in multiple bellies, neither analyzed by by multiple statistical indicators of responsiveness. At the same time, understanding the mechanisms behind the changes in TMG parameters caused by fatigue in both sexes, is also an area of research that needs further development.

Therefore, the primary objective of our study was to examine and compare the responsiveness of TMG parameters to QF peripheral fatigue of three muscle bellies (rectus femoris (RF), vastus lateralis (VL), and vastus medialis (VM)) by multiple statistical methods. A secondary objective was to examine whether there are differences between sex in the variation produced by fatigue in TMG parameters. Our hypotheses were: QF bellies have different responsiveness to peripheral fatigue; and the changes of TMG parameters are similar between males and females.

## MATERIALS & METHODS

### Study design

A single group pretest-posttest design was used, which involved repeated TMG and MVIC measures of the dominant lower limb QF before and after a fatigue protocol within the same session. Participants were physiotherapy students recruited by email using the University of Valencia Intranet. This study was conducted from April to July 2018. All measurements were carried out between 10 a.m. and 2 p.m in the clinical research laboratory of the Department of Physiotherapy (University of Valencia) at an ambient temperature 21–22 °C. An experienced examiner in the measurement techniques evaluated the participants. He was a physiotherapist who had used TMG and hand dynamometers both in research and in clinical practice for several years. Before participation, participants were informed of the study procedures and their possible associated risks. All of them provided written informed consent. This study was completed following the principles outlined in the Declaration of Helsinki and it was approved by the Ethics Committee of the University of Valencia (Spain) (H1523633864087).

### Participants

Thirty-nine recreational athletes were evaluated. All participants performed exercise 3 times per week and practiced activities such as running, swimming, cycling, or central strength training. The specific inclusion criteria were: (a) aged between 18 and 30 years, (b) not surgically
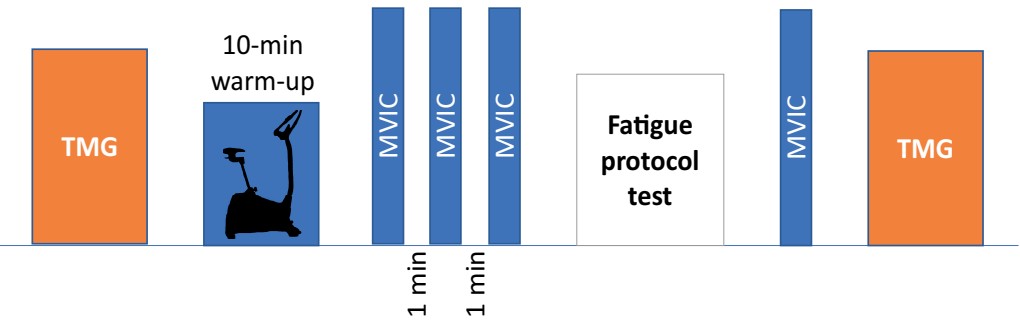

**Figure 1 Schematic representation of experimental procedures.** TMG, tensiomyography; MVIC, maximal voluntary isometric contraction.

operated on the lower limb, (c) without pain in the lower limb in the 2 months before data collection, and (d) performing physical exercise a minimum of 3 days per week. The exclusion criteria were: (a) practicing a specific sport as an amateur or professional, (b) contraindication to the use of electrodes due to injury or allergy to the adhesive, and (c) nontolerance to electrical stimulation.

## Procedures

Before starting the session, height was measured using a 1-millimeter sensitivity flexible tape measure, while body mass and body mass index (BMI) were assessed using a standardized body composition analyzer (Tanita BC 418 MA, Tanita Corp, Tokyo, Japan). Next, TMG parameters were measured and then, participants performed a warm-up, which consisted of 10 min cycling at comfortable speed (80 revolutions per minute) with low resistance and the performance of three submaximal isometric contractions of isometric knee extension (*Martins et al., 2017*). Following this, the MVIC test was performed. After the fatigue protocol, the order of the tests was reversed, and the strength test was performed first to reduce the time between MVIC and TMG tests in acute fatigue. A schematic representation of the experimental procedures is reported in the Fig. 1.

## Tensiomyography measurements

First, participants were placed supine and resting on the stretcher. The knee was placed at 120° of flexion (considering full extension at 180°), fixing such position with a triangular foam cushion (*García-García et al., 2016*; *Martín-San Agustín et al., 2020*). The area where the TMG sensor and electrodes were placed was shaved and cleaned with gauze and alcohol. The position of the sensor for each QF belly was determined using the anatomical criteria described in the literature (*Dahmane et al., 2005*; *Tous-Fajardo et al., 2010*; *Rey, Lago-Peñas & Lago-Ballesteros, 2012*). This position was marked with a permanent marker so that it would remain throughout the evaluation. The sensor was finally placed on this point perpendicularly to the thigh and the electrodes were placed at five cm distance from it, forming an imaginary straight line along the belly (Fig. 2).

The contractile properties of each belly were evaluated during an maximal elicited contractions with the TMG electro stimulator (TMG-100 System). Starting from 20 mA

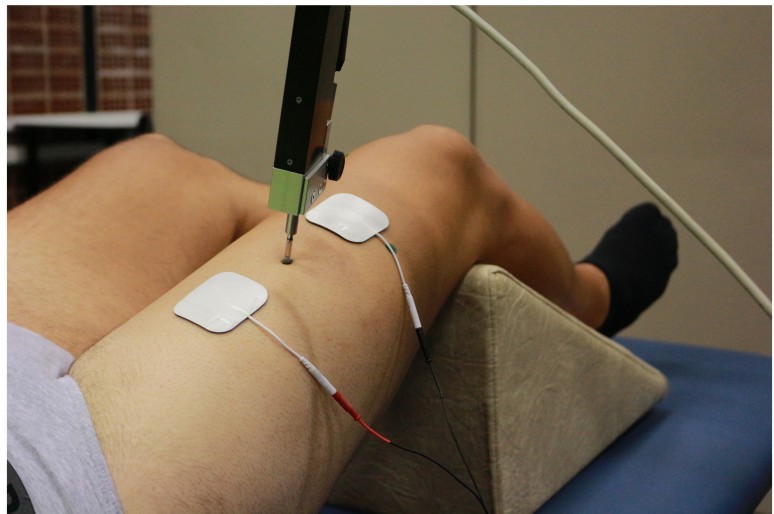

**Figure 2**  **Tensiomyographical measurement of rectus femoris.** Photo credit: Rodrigo Martín-San Agustín.

with 1ms pulses, each stimulation was increased by 10mA until achieving the maximum radial deformation (Dm) of the muscular belly. A time of 10s was left between stimuli to minimize fatigue or potentiation effects (*Krizaj, Simunic & Zagar, 2008*). Before data acquisition, a pilot test was done to verify the functioning of the TMG. For each belly, spatial and temporal parameters were measured: Dm, contraction time between 10 and 90% of the Dm (Tc), contraction velocity between 10 and 90% of the Dm (Vc), and contraction velocity of the first 10% of the Dm (V10). TMG has proven to be a method with a high relative [ICC for Dm (0.91–0.99), Tc (0.70–0.98), and VC > 0.95] and absolute (low coefficient of variations for Dm, Tc, and VC) reliability (*Martín-Rodríguez et al., 2017*; *Lohr et al., 2018*).

## Maximal voluntary isometric contraction test

MVIC of the QF was measured by a MicroFET2 handheld dynamometer (Hoggan Health Technologies Inc., Salt Lake City, UT). Participants were seated in an isokinetic dynamometer (Prima Plus, Easytech, Italy) with their torso and hips tied so they were stable, and with a 90° hip flexion. MVIC was evaluated in 90° knee flexion, considering 0° the complete extension (Fig. 3). MicroFET2 was fixed with a rigid belt perpendicular to the ankle five cm above the malleoli, with a pad between the tibia and the dynamometer to minimize the discomfort caused by the contact (*Hansen et al., 2015*).

After the warm-up, participants completed three MVIC for 5s, with a 60-second rest after each repetition. Through verbal stimuli, participants were instructed to exert and maintain the maximum effort during the session. MicroFET2 has proven to be a valid method to measure the MVIC of the QF with an excellent inter-examiner reliability (ICC: 0.93, 95% CI [0.83–0.97]) and a minimal detectable change (MDC) of 14.1 N*m (95% CI [9.23–22.01]) (*Hansen et al., 2015*).

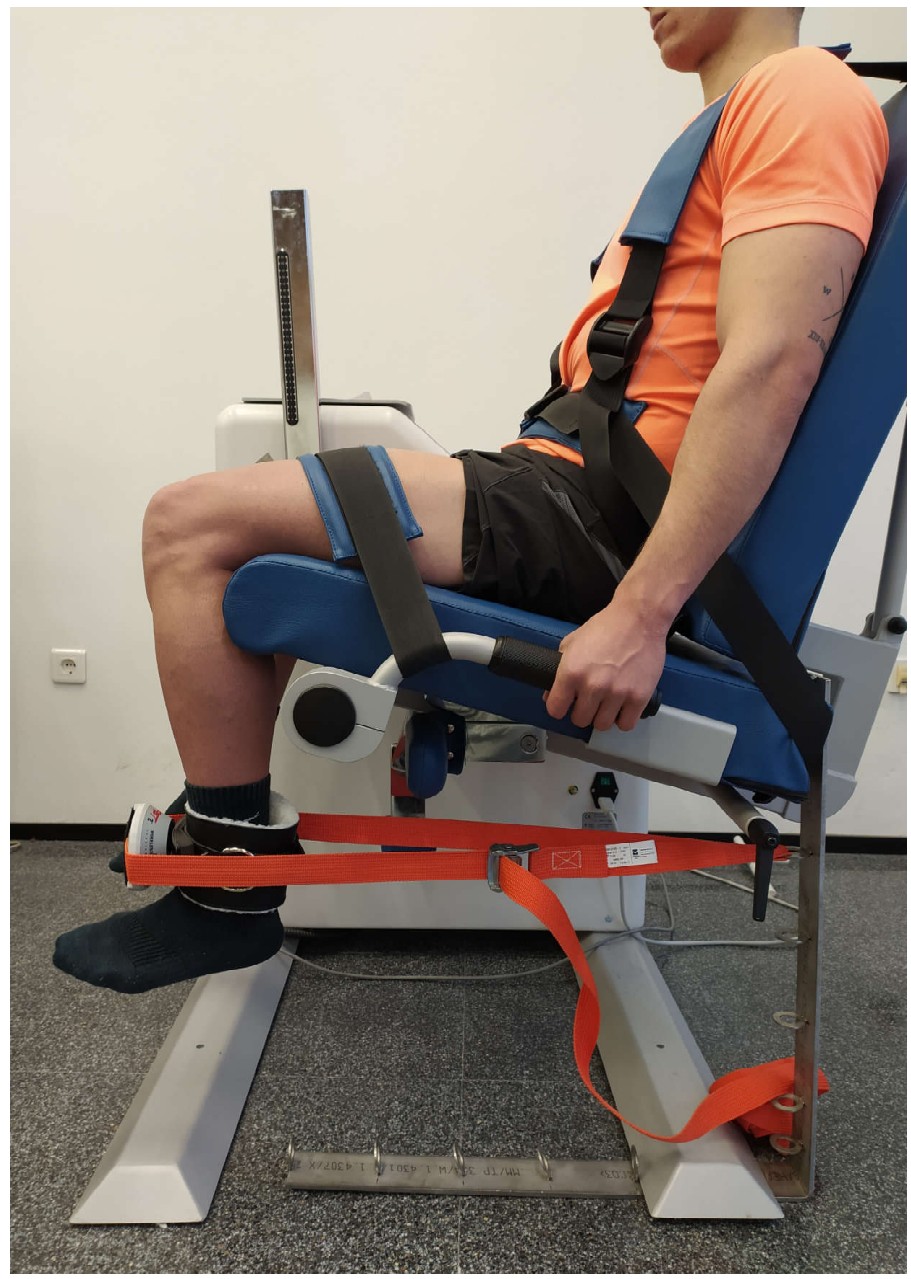

**Figure 3** **Maximal voluntary isometric contraction test for quadriceps femoris.** Photo credit: Rodrigo Martín-San Agustín.

## Fatigue protocol test

After performing the baseline measurements, participants were requested to implement a protocol based on a 60s fatiguing isometric contraction at 70% MVC (*Melchiorri & Rainoldi, 2011*). The experimental setup was the same as the one adopted during the MVIC test. The handheld dynamometer, previously set at 70% MVIC, was used to display the feedback (*Melchiorri & Rainoldi, 2011*). It was considered that the fatigue was achieved

when the reduction of the MVIC was higher than the upper limit of the MDC reported in a previous study (22.01 N*m) (*Hansen et al., 2015*).

## Statistical analysis

Baseline data were summarized as means and standard deviations (SD) for continuous variables and as absolute and relative frequencies for categorical variables. Variables were checked for normality with the Kolmogorov–Smirnov test and homogeneity of variances with Levene's test. A summary was also provided for participants with and without fatigued QF.

Paired t-tests were used to compare changes in the TMG parameters and MVIC within each sex group. These changes were also compared between sex groups by using non-paired t-tests.

Internal responsiveness was determined by the paired $t$-test and supplemented with an effect size statistic, as recommended by *Husted et al. (2000)* [30]. To reduce the probability of getting false positives, we increased the acceptance level from 0.05 to 0.01 for paired $t$-test because multiple comparisons were made on the same data set. Of the current effect size statistics we used the standardized response mean (SRM), which provides an estimate of the magnitude of change that is not influenced by sample size (*Navarro-Pujalte et al., 2018*). It was calculated as (MeanFollowup _ MeanBaseline)/Standard deviationFollowup-Baseline and the 95% confidence intervals were calculated using the bootstrapping estimation method. Values of 0.20, 0.50, and 0.80 or higher have been proposed in the literature (*Husted et al., 2000*) to represent small, moderate, and large responsiveness, respectively. Besides, we calculated the percentage of participants that exceeded MDC. This statistic examines the extent to which change score exceeds the amount of variability accounted by measurement error (*Pardasaney et al., 2012*), which is calculated as $SEM x 1.96 x \sqrt{2}$, where SEM is the standard error of measurement.

External responsiveness was determined by correlations, regression models, and receiver operating characteristic (ROC) curves (*Husted et al., 2000*). The external criterion for assessing the external responsiveness of the TMG tool was the magnitude of change in the MVIC.

We assumed that: (i) changes in the external criterion (MVIC) in participants with fatigue would be associated with changes in the TMG parameters; (ii) participants without fatigue would have the smallest change in the TMG parameters (and therefore change in these TMG parameters can be useful to classify participants' QF as fatigued or not fatigued). To test the first hypothesis, correlations and simple and multiple linear regression models were used. In the regression models the explanatory variable was the change of each TMG parameter while the response variable was the change in MVIC between before and after protocols. Each model was controlled by sex, and comparisons were carried out between the presence or absence of this control. Goodness-of-fit of the model was assessed by r2. To test the second hypothesis, we calculated the area under the ROC curve (AUC), which represents the probability that the measure of correctly classifying participants has (*Husted et al., 2000*). An AUC >0.70 was used as a generic benchmark to consider acceptable its discriminant ability (*Menaspà, Sassi & Impellizzeri, 2010*).

For sample size calculation, we selected the multiple regression as the main statistic of responsiveness because it allowed us to examine change relationships controlling by a covariate relevant in our study (sex). Regarding this statistic, we used the usual rule of thumb that 15 participants per predictor are needed for a reliable equation in multiple regression models (*Tabachnick & Fidell, 2007*). We recruited a minimum of 30 participants assuming a maximum of 2 explanatory variables (TMG parameter and sex). Statistical significance was set at $p < 0.05$. All analyses were performed using the Statistical Package for the Social Sciences software program (SPSS version 24.0; IBM SPSS, Chicago, IL, USA).

## RESULTS

### Participants' characteristics

Baseline characteristics of participants are listed in Table 1. A total of 35 (89.7%) participants achieved QF fatigue after the application of the fatigue protocol. They were 19 of 20 females (95%) and 16 of 19 males (84.2%). Participants with and without fatigue showed no significant differences ($p > 0.05$) in any of their baseline characteristics.

### Changes associated with the fatigue protocol

Participants with peripheral fatigue ($n = 35$) had a significant decrease (31.5%) on their MVIC after the fatigue protocol (from 203.3 N*m to 138.9 N*m). Table 2 shows that both sex groups had a similar pattern of change: males reduced 30.8% and females 32.1%. Table 2 also shows patterns of change by sex groups for TMG parameters of the RF, VL, and VM. All these parameters, except for the Tc of the RF and VM, had significant differences within but not between sex groups.

Figure 4 shows changes in TMG parameters for all participants with peripheral fatigue. All parameters, except for Tc, showed a significant difference ($p < 0.001$) for the three bellies of the QF. Dm's decrease ranged from 18.22% to 21.65%; Vc decreased from 15.62 to 22.20%, and V10 decreased from 14.80% to 23.77%.

### Internal and external responsiveness

Internal and external TMG responsiveness to fatigue of QF bellies is shown in Table 3. Internal responsiveness statistics suggest that all TMG parameters, except for Tc of RF and VM, showed large internal responsiveness (SRM > 0.8) among participants with QF fatigue. Dm and V10 in RF were the parameters in which most of the participants exceeded the MCD (91.3% and 97.1%, respectively). Only Dm, Vc, and V10 of the RF showed to be linearly associated with changes in the MVIC. After controlling by sex, adjusted models typically provided b coefficients and r2 with small variations regarding their respective unadjusted model (range 0.01 to 0.05). Consequently, Dm and V10 of RF were still statistically associated with b coefficients of 0.40 and 0.43, respectively. Moreover, the models of these parameters explained the 22% of the total variance.

The AUC analysis suggests that changes of several TM G parameters (Dm in RF and VL, Tc in VL, and V10 in RF and VM) were >0.70 and could discriminate between QF with and without fatigue. Also, the overlapping among their 95% CI suggests that none of these TMG parameters is superior to the others to discriminate fatigue.

**Table 1** Baseline characteristics of the participants in total and separated by fatigued condition.

| Baseline Characteristics | Total ($n = 39$) | Fatigued participants ($n = 35$) | Non-fatigued participants ($n = 4$) |
|---|---|---|---|
| Males/females, N (%) | 19 (48.7%)/20 (51.3%) | 16 (45.7%)/19 (54.3%) | 3 (75%)/1 (25%) |
| Age (years) | 22 (2) | 22 (2) | 21 (1) |
| Physical activity (minutes) | 316.5 (180.8) | 314.6 (186.7) | 332.5 (136.9) |
| Anthropometric | | | |
| Body mass (kg) | 67.37 (13.42) | 66.10 (11.12) | 78.55 (12.05) |
| Stature (cm) | 173.3 (9.50) | 172.5 (9.09) | 180.7 (11.24) |
| BMI (kg/m2) | 22.22 (2.72) | 22.02 (2.71) | 24 (2.53) |
| QF strength | | | |
| MVIC (N*m) | 207.56 (74.19) | 203.31 (75.82) | 244.72 (50.24) |
| Tensiomyography parameters | | | |
| Rectus femoris | | | |
| Dm (mm) | 10.26 (1.42) | 10.32 (1.44) | 9.76 (1.28) |
| Tc (ms) | 25.45 (4.04) | 25.69 (3.95) | 23.39 (4.84) |
| Vc (mm/s) | 327.96 (58.59) | 326.62 (69.76) | 339.70 (53.04) |
| V10 (mm/s) | 43.07 (5.32) | 43.08 (5.39) | 42.93 (5.33) |
| Vastus lateralis | | | |
| Dm (mm) | 5.74 (1.11) | 5.63 (0.94) | 6.64 (2.04) |
| Tc (ms) | 21.37 (3.02) | 21.54 (3.11) | 19.87 (1.35) |
| Vc (mm/s) | 217.78 (50.10) | 211.58 (39.81) | 271.95 (97.28) |
| V10 (mm/s) | 25.31 (5.18) | 24.73 (4.21) | 30.46 (9.98) |
| Vastus medialis | | | |
| Dm (mm) | 4.57 (0.85) | 4.52 (0.64) | 5.08 (2.01) |
| Tc (ms) | 19.60 (1.82) | 19.61 (1.90) | 19.48 (1.04) |
| Vc (mm/s) | 187.22 (33.12) | 185.08 (26.57) | 205.93 (73.31) |
| V10 (mm/s) | 23.22 (4.03) | 22.97 (2.89) | 25.37 (10.19) |

**Notes.**

Date represents mean and standard deviation unless otherwise noted.

BMI, body mass index; Dm, maximal radial displacement; Tc, contraction time; Vc, contraction velocity between 10–90% of the Dm; V10, contraction velocity of the first 10% of the Dm; QF, quadriceps femoris; MVIC, maximal voluntary isometric contraction.

# DISCUSSION

To our knowledge, this is the first study to evaluate the internal and external TMG responsiveness across a variety of QF muscle bellies to changes induced by peripheral fatigue. We found that TMG parameters Dm and V10 of the RF showed both internal and external responsiveness.

In our study, multiple statistical methods to evaluate the internal responsiveness (paired $t$-test and SRM) and external responsiveness (correlations, regression models and ROC) of the TMG were used, which is line with the recommendations of *Husted et al. (2000)*. In previous studies, most of these statistics have been used to evaluate only the TMG ability of change to fatigue (*García-Manso et al., 2011*; *De Paula Simola et al., 2015*). Thus, to the best of our knowledge, this is the first study to use several statistical methods to assess internal and external responsiveness. Furthermore, since most of the previous studies assessing fatigue by TMG have only evaluated isolated muscle bellies (*García-Manso et al., 2011*;
**Table 2  Differences within and between sex groups in the TMG parameters and MVIC after fatigue protocol.**

| Muscle | Males | | | | Females | | | |
|---|---|---|---|---|---|---|---|---|
| | Baseline | Fatigued | Differences | | Baseline | Fatigued | Differences | |
| | | | Mean (SD); *p* | % | | | Mean (SD); *p* | % |
| QF strength | | | | | | | | |
| MVIC (N*m) | 272.1 (51.0) | 187.3 (40.1) | 84.7 (37.8); <0.001 | 30.8 | 145.4 (30.7) | 98.1 (24.4) | 47.3 (22.3); <0.001 | 32.1 |
| Rectus femoris | | | | | | | | |
| Dm (mm) | 9.91 (1.66) | 7.46 (1.87) | 2.45 (1.27); <0.001 | 25.2 | 10.67 (1.16) | 8.71 (1.76) | 1.95 (1.13); <0.001 | 18.7 |
| Tc (ms) | 24.58 (4.25) | 24.52 (6.37) | 0.06 (3.28); 0.941 | 1.1 | 26.62 (3.52) | 27.63 (5.43) | −1.01 (4.42); 0.334 | 4.1 |
| Vc (mm/s) | 330.01 (78.95) | 250.71 (66.81) | 79.30 (48.65); <0.001 | 21.8 | 373.76 (39.15) | 256.21 (51.02) | 67.55 (42.26); <0.001 | 20.9 |
| V10 (mm/s) | 43.17 (6.55) | 32.78 (7.72) | 10.39 (5.35); <0.001 | 24.4 | 43.01 (4.37) | 33.01 (5.13) | 10.00 (4.20); <0.001 | 23.2 |
| Vastus lateralis | | | | | | | | |
| Dm (mm) | 5.47 (1.18) | 4.48 (0.76) | 0.99 (1.10); 0.003 | 20.5 | 5.78 (0.70) | 4.10 (1.15) | 1.68 (0.90); <0.001 | 29.5 |
| Tc (ms) | 21.69 (3.05) | 19.93 (4.31) | 1.76 (2.44); 0.011 | 8.6 | 21.42 (3.24) | 19.04 (1.88) | 2.38 (2.15); <0.001 | 10.4 |
| Vc (mm/s) | 203.67 (49.77) | 179.33 (66.24) | 24.35 (43.77); 0.042 | 12.8 | 218.24 (28.76) | 170.24 (37.41) | 48.00 (43.15); <0.001 | 20.9 |
| V10 (mm/s) | 24.28 (5.04) | 20.46 (6.78) | 3.82 (4.33); 0.003 | 17.3 | 25.10 (3.45) | 18.65 (4.66) | 6.45 (4.55); <0.001 | 25.3 |
| Vastus medialis | | | | | | | | |
| Dm (mm) | 4.69 (3.91) | 3.91 (0.78) | 0.78 (0.59); <0.001 | 16.3 | 4.37 (0.50) | 3.51 (0.69) | 0.86 (0.53); <0.001 | 19.8 |
| Tc (ms) | 20.25 (1.78) | 19.96 (2.66) | 0.28 (1.97); 0.573 | 1.4 | 19.07 (1.88) | 18.26 (1.88) | 0.81 (1.64); 0.045 | 3.9 |
| Vc (mm/s) | 186.06 (30.93) | 159.90 (25.72) | 29.16 (22.46); <0.001 | 14.9 | 184.26 (23.12) | 153.76 (29.26) | 30.50 (26.86); <0.001 | 16.2 |
| V10 (mm/s) | 23.76 (3.19) | 21.09 (3.95) | 2.67 (2.97); 0.003 | 11.2 | 22.31 (2.51) | 18.33 (3.40) | 3.98 (2.74); <0.001 | 17.8 |

**Notes.**

SD, standard deviation; Dm, maximal radial displacement; Tc, contraction time; Vc, contraction velocity between 10–90% of the Dm; V10, contraction velocity of the first 10% of the Dm; QF, quadriceps femoris; MVIC, maximal voluntary isometric contraction.

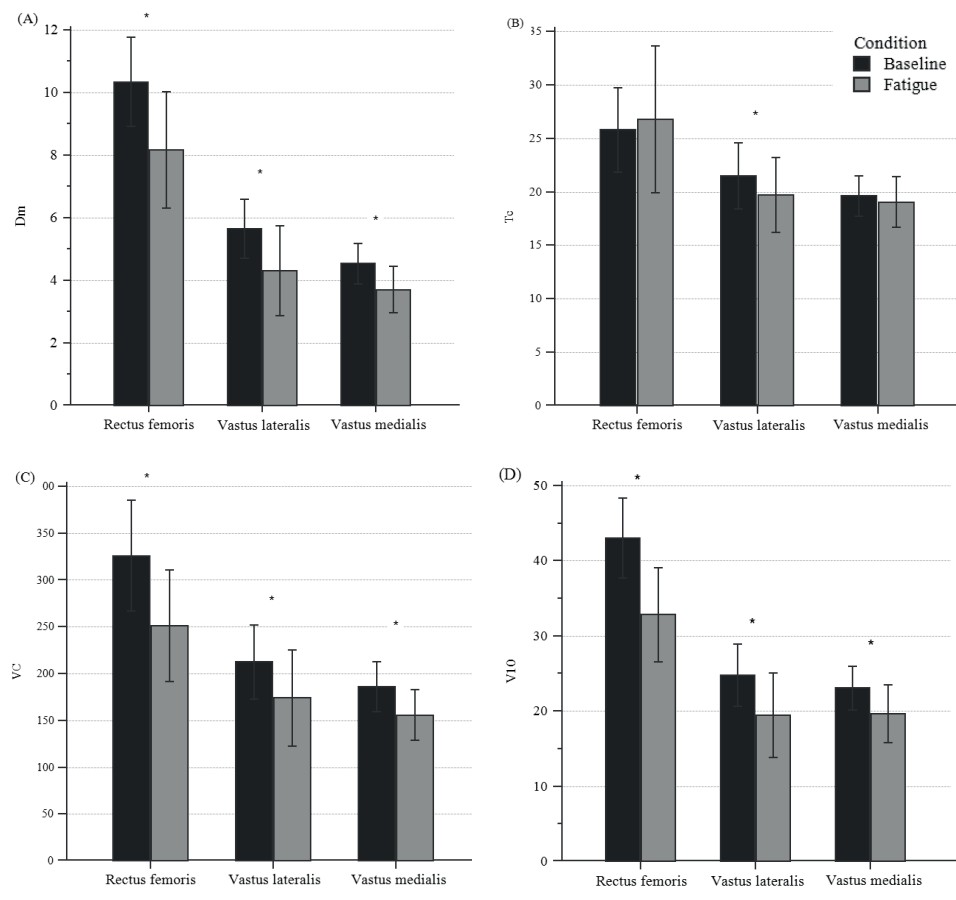

**Figure 4   Differences in TMG parameters of quadriceps bellies between pre- and post-fatigue in all participants.** (A) Differences in Dm, (B) in Tc, (C) in VC, and (D) in V10. *Significant differences set at $p < 0.01$; Specific $p$-values are shown in Table 3.

*Hunter et al., 2012*; *De Paula Simola et al., 2015*; *De Paula Simola et al., 2016*; *Giovanelli et al., 2016*; *Macgregor et al., 2016*; *Raeder et al., 2016*; *Wiewelhove et al., 2017*), our study presents novel findings in the evaluation of TMG across multiple muscle bellies.

Regarding the internal responsiveness, large and negative SRM of the TMG parameters were found in most of the muscle bellies. Overall, our results are consistent with previous studies that induced peripheral and central QF fatigue (i.e., selective QF fatigue or caused in the entire lower limb musculature). Therefore, the reduction of RF TMG parameters is consistent with previous studies using peripheral (*Carrasco et al., 2011*) or central fatigue (*De Paula Simola et al., 2015*), finding them reductions in Dm, VC, or V10 after fatigue due to cycling or strengthening. On the other hand, the changes in VL and VM are also consistent with studies using central fatigue caused by strengthening programs (*De Paula Simola et al., 2016*; *Raeder et al., 2016*). In addition, Dm results showed consistence with other studies that induced peripheral fatigue in muscles such as the biceps brachii (*Hunter et al., 2012*; *García-Manso et al., 2012*) or the gastrocnemius medialis (*Macgregor et al., 2016*). These findings could be explained by changes in the pH (*Hunter et al., 2009*) and

Martin-San Agustín et al. (2020), *PeerJ*, DOI 10.7717/peerj.8674

**Table 3** Responsiveness statistics for the TMG parameters.

| Muscle | Internal responsiveness | | | External responsiveness | | | |
|---|---|---|---|---|---|---|---|
| | Paired *t*-test (*p*) | SRM (95% CI) | % MCD | Correlation method (Pearson's r and 95% CI); *p* | Linear regression method[a] | | AUC (95% CI) |
| | | | | | b(SE); *p* | r2 | |
| Rectus femoris | | | | | | | |
| Dm (mm) | 0.001 | −1.83 (−2.31; −1.47) | 91.3 | 0.42 (0.12; 0.65); 0.004 | 0.40 (0.14); 0.007 | 0.22 | 0.73 (0.57; 0.86) |
| Tc (ms) | 0.439 | 0.13 (−0.24; 0.39) | 15.9 | 0.10 (−0.22; 0.40); 0.276 | 0.14 (0.15); 0.363 | 0.06 | 0.62 (0.45; 0.77) |
| Vc (mm/s) | 0.001 | −1.65 (−1.98; −1.30) | 79.7 | 0.33 (0.02; 0.58); 0.020 | 0.26 (0.13); 0.052 | 0.13 | 0.59 (0.42; 0.74) |
| V10 (mm/s) | 0.001 | −2.20 (−2.65; −1.78) | 97.1 | 0.45 (0.15; 0.67); 0.002 | 0.43 (0.15); 0.006 | 0.22 | 0.73 (0.57; 0.86) |
| Vastus lateralis | | | | | | | |
| Dm (mm) | 0.001 | −1.33 (−1.74; −0.82) | 79.7 | 0.18 (−0.14; 0.47); 0.133 | 0.10 (0.12); 0.403 | 0.05 | 0.81 (0.65; 0.92) |
| Tc (ms) | 0.001 | −0.87 (−1.27; −0.41) | 65.2 | 0.12 (−0.12; 0.48); 0.111 | 0.23 (0.19); 0.238 | 0.07 | 0.92 (0.79; 0.98) |
| Vc (mm/s) | 0.001 | −0.86 (−1.21; −0.46) | 43.5 | 0.09 (−0.23; 0.39); 0.298 | 0.03 (0.11); 0.782 | 0.04 | 0.55 (0.39; 0.71) |
| V10 (mm/s) | 0.001 | −1.17 (−1.56; −0.71) | 68.1 | 0.12 (−0.20; 0.42); 0.224 | 0.06 (0.12); 0.638 | 0.04 | 0.67 (0.50; 0.81) |
| Vastus medialis | | | | | | | |
| Dm (mm) | 0.001 | −1.46 (−1.84; −1.07) | 76.8 | 0.12 (−0.21; 0.42); 0.116 | 0.09 (0.20); 0.643 | 0.04 | 0.65 (0.48; 0.79) |
| Tc (ms) | 0.069 | −0.34 (−0.72; 0.02) | 42 | −0.14 (−0.43; 0.18); 0.200 | −0.28 (0.28); 0.331 | 0.06 | 0.52 (0.36; 0.68) |
| Vc (mm/s) | 0.001 | −1.17 (−1.50; −0.79) | 68.1 | 0.17 (−0.15; 0.46); 0.143 | 0.17 (0.19); 0.364 | 0.06 | 0.68 (0.52; 0.82) |
| V10 (mm/s) | 0.001 | −1.14 (−1.47; −0.76) | 71 | 0.26 (−0.06; 0.53); 0.054 | 0.25 (0.19); 0.194 | 0.08 | 0.76 (0.60; 0.88) |

Notes.

SRM, standardized response mean; CI, confidence interval; MCD, minimal detectable change; SE, standard error; AUC, area under curve; Dm, maximal radial displacement; Tc, contraction time; Vc, contraction velocity between 10–90% of the Dm; V10, contraction velocity of the first 10% of the Dm.

[a] Adjusted by sex.

in different cellular molecules (e.g., Na+ or K+) (*Brody et al., 1991*), which cause damage in the sarcolemma and the reduction of the electrical stimulus, with a possible decrease in muscle displacement.

This study showed that Dm and V10 of RF had an acceptable external responsiveness in relation to our external criterion, namely changes in the strength evidenced by MVIC. As reflected by the regression coefficients, there was a moderate relationship between the amount of change in TMG parameters and strength scores. This relationship is consistent with a previous study using central fatigue (*De Paula Simola et al., 2015*). Furthermore, Dm and V10 were relevant according to sex, which can be explained by the fact that our sample showed similar change magnitudes in both TMG parameters and strength scores.

The fatigue protocol used in this study was highly effective (most of the QF showed fatigue). Males and females had similar strength change scores (Table 2). Previous studies reported different strength change scores between sexes when intensities between 25–50% of MVIC were used (*Clark et al., 2005*; *Ansdell et al., 2017*). In our study, an intensity of 70% of MVIC was used, suggesting that as the contraction intensity increase, the sex differences in muscle fatigue decrease , (*Hunter, 2014*). Therefore, future investigations should examine whether sex differences in strength changes are detected by sex differences in the TMG changes.

Our present study also showed that TMG has discriminative ability to classify the participants' QF as having fatigue or not after the application of the protocol. Dm and V10 of the RF also were two of the four parameters with this discriminative ability. This finding is partially consistent with previous studies (*Wiewelhove et al., 2017*), who examined AUC of RF after central fatigue in elite young athletes. Nevertheless, while AUC values of V10 shown in this study was similar to their results, AUC values of Dm was higher than previously published (*Wiewelhove et al., 2017*). Differences may be explained by the different type of fatigue (central fatigue caused by several training sessions of high-intensity interval training vs peripheral fatigue by an MVIC test) or by the athletes' training background (junior tennis players vs recreational athletes). Other parameters with that discriminative ability were Dm and Tc of VL, and V10 of VM. Since this ability was not previously analyzed in these muscle bellies (VL and VM), results of the actual study supplements earlier findingswhich have only evaluated AUC for external responsiveness of the TMG in RF (*Wiewelhove et al., 2017*) and it provides evidence to expand the application of the TMG to discriminate fatigue.

Actual study has several limitations. First, we used a fatigue protocol based on MVIC, which induces peripheral fatigue. Therefore, our findings would be limited to be extrapolated to others fatigue situations (e.g., concentric contractions). Second, our study was conducted with recreational athletes (i.e., anyone participating in an aerobic or athletic activity at least three times per week) (*Heinert et al., 2008*). Since the contractile properties of the muscle are conditioned by the type of exercise performed (*Loturco et al., 2015*), future research should compare our results with findings from athletes of different sports.

Our study found that most of the TMG parameters showed an acceptable internal responsiveness of QF peripheral fatigue evidenced by a reduction of the MVIC. In contrast,

only Dm and V10 of RF showed external responsiveness. Therefore, our study illustrates that the use of only internal or external responsiveness may lead to incomplete conclusions (*Husted et al., 2000*). In this way, professionals should use both, as recommended by Husted (*Husted et al., 2000*).

This study showed that Dm and V10 of RF measured by TMG were both internally and externally responsive to changes between before and after a peripheral fatigue protocol. Since the QF is the main strength contributor during cycling (*Raasch et al., 1997*) or running (*Montgomery, Pink & Perry, 1994*), the fatigue evaluation after an effort is essential to manage recovery of the athlete and the intensity of subsequent training sessions. Thus, clinicians and trainers should be able to direct the fatigue evaluations without making new efforts with TMG, taking into consideration Dm and V10 parameters in RF to discriminate the presence of peripheral fatigue and the magnitude of the strength changes and, in this way, be able to regulate training loads (e.g., in the presence of peripheral fatigue, decrease intensity or activities that involve the QF).

## CONCLUSIONS

According to the results, it can be concluded about positive responsiveness of the TMG in peripheral fatigue of the QF, demonstrating that the Dm and V10 parameters of the RF present acceptable responsiveness to fatigue. Therefore, by using the TMG, it is possible to determine whether the QF shows peripheral fatigue or not, and to relate changes in the parameters with the reduction of strength. Thus, clinicians and trainers should be able to direct the fatigue evaluations without making new efforts with TMG, facilitating the regulation of training loads. Finally, future studies should examine the responsiveness of TMG to other types of fatigue and in other sports.

## ACKNOWLEDGEMENTS

The authors thank the volunteers for their cooperation during the course of this study.

### Funding
The authors received no funding for this work.

### Competing Interests
The authors declare there are no competing interests.

### Author Contributions
- Rodrigo Martín-San Agustín conceived and designed the experiments, performed the experiments, analyzed the data, authored or reviewed drafts of the paper, and approved the final draft.
- Francesc Medina-Mirapeix conceived and designed the experiments, analyzed the data, prepared figures and/or tables, authored or reviewed drafts of the paper, and approved the final draft.

- José Casaña-Granell and Josep C. Benítez-Martínez conceived and designed the experiments, performed the experiments, authored or reviewed drafts of the paper, and approved the final draft.
- José A. García-Vidal analyzed the data, prepared figures and/or tables, and approved the final draft.
- Carmen Lillo-Navarro analyzed the data, prepared figures and/or tables, authored or reviewed drafts of the paper, and approved the final draft.

## Human Ethics

The following information was supplied relating to ethical approvals (i.e., approving body and any reference numbers):

The University of Valencia granted ethical approval to carry out the study within its facilities (Ethical Application Ref: H1523633864097).

## Data Availability

The raw measurements are available in the Supplemental Files.

## Supplemental Information

Supplemental information for this article can be found online at http://dx.doi.org/10.7717/peerj.8674#supplemental-information.

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
