# Peer review of "Tensiomyographical responsiveness to peripheral fatigue in quadriceps femoris"

_PeerJ, doi:10.7717/peerj.8674_

## Round 0.1 · original submission · Major Revisions

Expert reviewers carefully evaluated your manuscript. Authors need to take care of formatting and presentation style etc. and experimental concerns noted by the reviewers. They identifed problems regarding the quality of English and a number of methodological questions also need to be clarified. Thus, you and your research team need to really revise your paper with this in mind.

·

Basic reporting

Manuscript entitled as: Tensiomyographical responsiveness to local fatigue in quadriceps femoris, have clear scientific idea and well organized research/experimental protocol. Results of this study are actual and applicable in sports and rehabilitation practice. According to my opinion, the article reach journal standards for publishing, but after minor technical corrections (marked in PDF format of the manuscript).

Experimental design

Experimental design is organized and realized on proper way, with well and adequately verify by relevant references.

Validity of the findings

Results of the study are clear and confirmed by applied statistical analyses.

Additional comments

Manuscript is written acceptable well and clear. Measurement protocol and statistical analyses was done on proper way. After suggested minor technical text improvement, manuscript should be accepted for publication.

Reviewer 2 ·

Basic reporting

I congratulate authors for an important contribution in understanding relation between TMG and fatigue. Although, i have several issues that needs to be addressed:

1. It would be better to operate with peripheral and central fatigue, not local and general;
2. Fourth section in Introduction is less clear to a reader. Furthermore, the reference Husted et al. (2000) is not reflecting its purpose. Husted did not publish TMG article;
3. Instead of submaximal involuntary contractions put maximal elicited contractions. It would be clearer.
4. Use equation editor to insert equations.
5. The fourth paragraph of Statistics should start as: We assumed... Furthermore, was the external standard the same as external criterion (MVIC) ?
6. Use abbreviations consistently after they are defined and define abbreviations before their first use (e.g. Abstract Tensiomyography -> TMG)
7. Commas are missing or they are some extra (e.g. before respectively; ,, in discussion)
8. Figure and Table with capital letter.
9. Avoid unnecessery abbreviations in Tables (VL, VM, RF and TMG could be written down). If there is no significnace remove data about differences. use only p instead of p.value
10. Figure 2 (no colours, no horizontal lines, only one legend)

Experimental design

1. Why did you use multiple t-test instead of 2-way ANOVA (or GLM)? What was the rationale for this? Did you compensate P-value for multiple comparisons?
2. Was placement of sensor in line with the most recent finding of Šimunič (2019) Journal of Biomechanics?
3. Move in Methods->procedures statement from Methods-Statistics (It was considered that the fatigue was achieved when the reduction of the MVIC...)

Validity of the findings

1. Figure 2: Why did you report only those fatigued and not all of them? What was the rationale for such an approach?
2. How could you interpret Tc? And how decrease of Tc after fatigue? You found the opposite as Hunter et al. (2012). Any comments? Was your fatiguing protocol sufficient to fatigue fast twitch MUs?
3. Could you report (somewhere in text) what was the average time until fatigue. This would be additional information that would allow interpretation of outcomes changes.
4. Last sentence in Results: why did you at this point expect that TMG parameters must be discrimination tool with 100% Se and 100% Sp? In Discussion and Conclusions you've stated that it could be.
5. Last sentence in Discussion: This could be applied only to 70% isometric exercise. Soften it up.

Reviewer 3 ·

Basic reporting

see below

Experimental design

see below

Validity of the findings

see below

Additional comments

ARTICLE REVIEW

TITLE: Tensiomyographical responsiveness to local fatigue in quadriceps femoris

Human participant/human tissue checks

Have you checked the authors ethical approval statement?
Yes, I checked.

Does the study meet our article requirements?
Yes, it does.

Has identifiable info been removed from all files?
Yes, it has.

Were the experiments necessary and ethical?
Yes, they were.


STRUCTURE

1. Basic Reporting

Although this is a well-constructed study, the English language needs to be reviewed. The introduction and discussion sections are well structured and follow a logical sequence. The materials & methods section needs to be better structured in order to enable the reader to understand the sequence of the experimental procedures. The authors support the information with relevant literature in the field. The structure of the manuscript conforms to PeerJ standards. The figure presented by the authors is relevant, of high quality, and is properly labelled and described. However, as I detail in the general comments, more figures are needed for a better understanding of the experimental procedures. The authors supplied the raw data.

2. Experimental Design

The present study is an original primary research, which falls within the scope of the PeerJ journal. The authors define the research question, which seems to be relevant in the field of fatigue evaluation, and clearly state how their approach to the problem fills an identified knowledge gap. The experimental procedures were performed to a high technical and ethical standard. The methods section needs to be further developed in order to better elucidate the reader.


3. Validity of the findings

The rationale and benefit to the literature was clearly stated by the authors, presenting both the strengths and limitations of the study. The data collected was robust and the analyses performed were appropriate to the research question. The conclusions were well stated, linked to the original research question and limited to the obtained results.

4. General comments

The aims of the present study were to examine and compare the tensiomyographical (TMG) responsiveness to quadriceps femoris (QF) fatigue by multiple statistical methods and to analyze differences between sex in the variation produced by fatigue in TMG parameters. The main findings were that TMG parameters, such as maximum radial deformation (Dm) and contraction velocity of the first 10% of the Dm (V10) of the rectus femoris (RF), showed both internal and external responsiveness, and both sexes exhibited similar strength change scores. This is a well-constructed study (valid and reliable methodological design), with novel practical findings related to fatigue measurement. In this regard, the authors should be congratulated for this interesting manuscript. The results presented here would be of interest to the readers of this Journal. However, there are some major comments and concerns that I will detail below in order to improve the quality of the manuscript, including spelling and grammar.

Abstract

- Line 17: Please rewrite the sentence “Fatigue influences athletic performance or increased risk of injury in sports…” to “Fatigue influences athletic performance and can also increase the risk of injury in sports…”

- Line 20: Please, change “Even so” to “However”

- Line 21-23: Please rewrite “The aim of this study was to examine and compare the tensiomyographical responsiveness to quadriceps femoris (QF) fatigue by multiple statistical methods.” to “Thus, the aim of the present study was twofold: (i) to examine and compare the tensiomyographical responsiveness to quadriceps femoris (QF) fatigue by multiple statistical methods and (ii) to analyze sex differences in the variation produced by fatigue in TMG parameters.”

- Line 33: Please, change “R2” to “r2”

- Line 24: Please, specify how many males and females participated in the study and the average age ± SD

- In results, I recommend the authors to report the statistical significance (e.g., p < 0.05)

- I suggest the authors to add a conclusion section with the last sentence (“Since the QF is the main strength contributor during multiple physical activities, clinicians and trainers will be able to discriminate the presence of fatigue and the magnitude of changes in the QF strength by TMG evaluation.”)

Introduction

- Line 47-49: Please rewrite the sentence “Since fatigue influences athletic performance (Thorlund et al., 2008; Ditroilo et al., 2011) or increased risk of injury in sports (Zebis et al., 2011; Liederbach et al., 2014), its study has been of interest.” to “Since fatigue influences athletic performance (Thorlund et al., 2008; Ditroilo et al., 2011) and can also increase the risk of injury in sports (Zebis et al., 2011; Liederbach et al., 2014), its study has been of interest.”

- Line 56-62: The sentence is too long and confusing. I suggest the authors shortening the sentence and be more specific.

- Line 87: Please rewrite “However, the external responsiveness of TMG has not been yet assessed for local fatigue…” to “However, to the best of our knowledge, the external responsiveness of TMG has not been yet assessed for local fatigue…”

- Line 89-91: I suggest the authors to rewrite the sentence for clarity. For example: “Furthermore, to our knowledge, TMG responsiveness has not been simultaneously evaluated in multiple muscle bellies, neither analyzed by multiple statistical indicators of responsiveness. At the same time, understanding the mechanisms behind the changes in TMG parameters caused by fatigue in both sexes, is also an area of research that needs further development.”

Materials and Methods

- Before the participants subsection, please consider to add a study design subsection, referring the following: type of design; a brief description of the subjects; a brief description of the experimental procedures (i.e., the study time period, the place where the experimental procedures were performed, as well as their environmental conditions, like temperature and humidity); the role of the evaluators during the tests (their experience administering the tests); the number of sessions needed to conduct the evaluations; and finally the ethical approval.

- I suggest the authors to rewrite the participants subsection, specifying the relevant information (e.g., inclusion and exclusion criteria) and removing what was already been referred in the study design subsection.

- Line 103: What the authors mean with “recreational athletes”? Please, specify with some examples of sports.

- Line 103-104: “…that was conducted from April to July 2018”. Please, move this information to the study design subsection

- In the procedure’s subsection, sometimes is hard to understand the sequence of the experimental design. Thus, for a better understanding, I suggest the authors to add a schematic representation of the procedures (e.g., see https://doi.org/10.3389/fphys.2018.01503 or https://doi.org/10.1371/journal.pone.0091754), and if possible, to insert new subsections between the following lines: 114-121 (e.g., Procedures); 122-144 (e.g., Tensiomyography measurements); 145-156 (e.g., Maximal voluntary isometric contraction test); 157-161 (e.g., Fatigue protocol test).

- Line 115: Please, specify the instrument used to measure the subject’s height.

- Line 115: Please rewrite “height was measured using a tape measure and body mass and body mass index (BMI) were registered…” to “height was measured using a tape measure, while body mass and body mass index (BMI) were assessed…”.

- Line 117-121: Please, rewrite these lines for clarity. The reader should easily understand the sequence of the procedures. What was the first procedure? It was the warm-up? If so, describe it. What was the second procedure? For a clear understanding, I suggest the authors to describe the procedures with a logical sequence.

- Line 119: “…at comfortable speed with low resistance…”. I suggest the authors to detail both the speed and the resistance, for replicate reasons.

- Line 121: “…the order of the tests was reversed…”. I recommend the authors to specify which tests were reversed and why they did that. What was the purpose? This is not clear for the reader.

- Line 144: “10% of the Dm (V10).”. After this line, I suggest the authors to present the relative and absolute reliability of the TMG parameters, through the intraclass correlation coefficient and the coefficient of variation, respectively. It is important to inform the reader about the consistency of the measurements performed.

- Line 149-151: For a better understanding, I suggest the authors to insert a figure demonstrating the procedures described.

Line 154-155: “After the warm-up, participants completed three MVIC for 5s, with a 60-second rest after each repetition.”. As mentioned above for the TMG parameters, here I also recommend the authors to present the relative and absolute reliability of the MVIC measurements.

- Line 163: Please, remove the end point in “Statistical analysis.”

- Line 173-175: As mentioned above, please consider moving the definition of internal responsiveness to the introduction section (see https://doi.org/10.1111/jan.12898). Include it in line 82, after the sentence “…reference measure or regression models) (Husted et al., 2000).”

- Line 175: “Of the current effect size statistics…”. Since the most appropriate effect size for calculating responsiveness remains controversial, why did you use the standardized response mean (SRM), instead of the Cohen’s d effect size (ES), or the standardized effect size (SES)? None of the formulas are influenced by the sample size. If the authors consider it appropriate, they can present the results of the three effect sizes and analyze possible differences between them (see https://doi.org/10.1111/jan.12898).

- Line 185-187: As mentioned above, please consider moving the definition of external responsiveness to the introduction section (see https://doi.org/10.1111/jan.12898). Include it in line 82, after the sentence “…reference measure or regression models) (Husted et al., 2000).”

- Line 198: Please, change “R2” to “r2”.

- Line 207: Before the sentence “All analyses were performed…”, please consider including the following sentence: “Statistical significance was set at p < 0.05.”

Results

- Please, put in italic the p-value

- Line 240: Please, change “97.1” to “97.1%”

- Line 242: Please, change “R2” to “r2”.

- In the tables, please also put in italic the p-value and change “R2” to “r2”.


Discussion

- Line 260-262: Please, rewrite “We used multiple statistical methods to evaluate the internal responsiveness (paired t-test and SRM) and external responsiveness (correlations, regression models and ROC) of the TMG, as recommended by Husted et al. (2000)” to “In our study, multiple statistical methods to evaluate the internal responsiveness (paired t-test and SRM) and external responsiveness (correlations, regression models and ROC) of the TMG were used, which is line with the recommendations of Husted et al. (2000).”

- Line 264-266: Please, rewrite “Thus, one strength of our study is that, as far as we know,, this is the first study evaluating various statistics from internal and external responsiveness.” to “Thus, to the best of our knowledge, this is the first study to use several statistical methods to assess internal and external responsiveness.”

- Line 265-269: Please, rewrite “An additional strength is that we evaluated TMG across multiple muscle bellies within the same study. Most of previous studies assessing fatigue by TMG have only evaluated isolated muscle bellies (García-Manso et al., 2011; Hunter et al., 2012; de Paula Simola et al., 2015, 2016; Giovanelli et al., 2016; Macgregor et al., 2016; Raeder et al., 2016; Wiewelhove et al., 2017)” to “Furthermore, since most of the previous studies assessing fatigue by TMG have only evaluated isolated muscle bellies (García-Manso et al., 2011; Hunter et al., 2012; de Paula Simola et al., 2015, 2016; Giovanelli et al., 2016; Macgregor et al., 2016; Raeder et al., 2016; Wiewelhove et al., 2017), our study presents novel findings in the evaluation of TMG across multiple muscle bellies.”

- Line 270-271: “Regarding the internal responsiveness, large and negative SRM of the TMG parameters were found in most of the muscle bellies.” Could the authors present a possible explanation for this occurrence?

- Line 275: Please, change “Changes” to “changes”

- Line 277: Please, rewrite “…other studies that induced local fatigue in other muscles…” to “…other studies that induced local fatigue in muscles…”

- When the authors state that their results are consistent with previous findings, I suggest them to provide additional explanations. The reader must be informed about these consistencies.

- Line 279: Please, change “finding” to “findings”

- Line 287: Please, change “according sex” to “according to sex”

- Line 290-294: Please rewrite: “According to previous studies (Clark et al., 2005; Lee et al., 2017; Ansdell et al., 2017), this was an unexpected finding, which was probably due to the use of higher intensities in our study (70% MVIC) compared to the strengths used in previous studies (Clark et al., 2005; Ansdell et al., 2017), since sex differences in muscle fatigue decrease as the contraction intensity increases (Hunter, 2014).” to “Previous studies reported different strength change scores between sexes when intensities between 25-50% of MVIC were used (Clark et al., 2005; Ansdell et al., 2017). In our study, an intensity of 70% of MVIC was used, suggesting that as the contraction intensity increase, the sex differences in muscle fatigue decrease (Hunter, 2014).”

- Line 299: Please, change “authors” to “studies”

- Line 300-301: “Nevertheless, while AUC of V10 shown in our study was similar to their results, AUC of Dm was higher than theirs (Wiewelhove et al., 2017).”. Please, rewrite this sentence for clarity.

- Line 310: Please, change “had” to “has”

- Line 310-311: Why the authors consider a limitation using an effective protocol to induce local fatigue? This could be a future study suggestion, but not a limitation. Please explain for clarity

- Line 319: Please, change “evinced” to “evidenced”

- Line 319: Please, specify "them". Which variables showed external responsiveness?

- Line 328-329: Please, change “taking in consideration” to “taking into consideration”

Conclusions

- Please, rewrite this section. I recommend the authors starting with the main results of the study. Next, I recommend mentioning the practical applications of the findings, and finally, I suggest presenting some future research directions (see lines 327-331).

---

## Round 0.2 · accepted · Accept

Three reviewers were satisfied with the changes implemented in the manuscript. Congratulations!

·

Basic reporting

Authors do change all what was asked in the first review. In line 328 there is one extra comma as a simply typing error, which should be deleted.

According to my opinion manuscript is in full ready to be accepted for publication. Good work.

Experimental design

Design of the study is according with positive scientific practice.

Validity of the findings

Findings are at high level of applicability in sports and rehabilitation practice.

Additional comments

Excellent work.

Reviewer 2 ·

Basic reporting

no comment

Experimental design

no comment

Validity of the findings

no comment

Additional comments

Authors successfully corrected the manuscript to meet all reviewers comments raised.

Reviewer 3 ·

Basic reporting

The authors made a good job per instructions.

Experimental design

The authors made a good job per instructions.

Validity of the findings

The authors made a good job per instructions.

Additional comments

The authors made a good job per instructions.